# SDNET2021: Annotated NDE Dataset for Subsurface Structural Defects Detection in Concrete Bridge Decks

Eberechi Ichi *, Faezeh Jafari and Sattar Dorafshan

Department of Civil Engineering, College of Engineering, University of North Dakota, 243 Centennial Drive Stop 8115, Grand Forks, ND 58202-8115, USA
* Correspondence: eberechi.ichi@und.edu

**Abstract:** Annotated datasets play a significant role in developing advanced Artificial Intelligence (AI) models that can detect bridge structure defects autonomously. Most defect datasets contain visual images of surface defects; however, subsurface defect data such as delamination which are critical for effective bridge deck evaluations are typically rare or limited to laboratory specimens. Three Non-Destructive Evaluation (NDE) methods (Infrared Thermography (IRT), Impact Echo (IE), and Ground Penetrating Radar (GPR)) were used for concrete delamination detection and reinforcement corrosion detection. The authors have developed a unique NDE dataset, Structural Defect Network 2021 (SDNET2021), which consists of IRT, IE, and GPR data collected from five in-service reinforced concrete bridge decks. A delamination survey map locating the areas, extent and classes of delamination served as the ground truth for annotating IRT, IE and GPR field tests' data in this study. The IRT were processed to create an ortho-mosaic maps for each deck and were aligned with the ground truth maps using image registration, affine transformation, image binarization, morphological operations, connected components and region props techniques to execute a semi-automatic pixel–wise annotation. Conventional methods such as Fast Fourier transform (FFT)/peak frequency and B-Scan were used for preliminary analysis for the IE and GPR signal data respectively. The quality of NDE data was verified using conventional Image Quality Assessment (IQA) techniques. SDNET2021 dataset consists of 557 delaminated and 1379 sound IE signals, 214,943 delaminated and 448,159 sound GPR signals, and about 1,718,083 delaminated and 2,862,597 sound IRT pixels. SDNET2021 addresses one of the major gaps in benchmarking, developing, training, and testing advanced deep learning models for concrete bridge evaluation by providing a publicly available annotated and validated NDE dataset.

**Keywords:** bridge; non-destructive evaluation; unmanned aerial system (UAS); impact echo (IE); ground penetrating radar (GPR); infrared thermography (IRT); data validation; fast fourier transformation; B-scan; artificial intelligence (AI)

## 1. Introduction

Inspectors are faced with the challenge of periodically inspecting over four billion square meters of reinforced concrete bridge deck; therefore, the demand for automating infrastructure construction, inspection, and planning using artificial intelligence (AI) has increased [1,2].

A comprehensive bridge deck evaluation requires the detection of surface and subsurface defects. Supervised deep learning networks, such as convolution neural networks (CNNs), which requires ground truth for validation, have yielded the most promising results among the AI methods for bridge deck evaluation due to their high accuracy [1,2]. AI-enabled bridge deck evaluation requires annotated realistic non-destructive evaluation (NDE) datasets. Datasets for surface defects, such as concrete cracks, where images are labeled to a class, such as cracked or intact, exist. However, NDE datasets are rare for adequate bridge evaluation using AI models [3–5].

Available datasets for bridge evaluations are typically limited to surface defects such as concrete cracks. Some were annotated at the pixel level, where each pixel was assigned to a class [6], while others used a bounding box to annotate surface defects [7]. McLaughlin et al. (2019) developed a dataset of 500 IRT images from four reinforced concrete bridges. All images were collected using a 512 × 640 pixel resolution: 261 with potential delamination and 239 without delamination. The images were annotated using a semantic pixel-wise method and classified as sound or delaminated; however, the annotations were not based on actual in-service concrete delamination [5]. Dorafshan et al. (2020a&b) presented an annotated impact echo dataset (IE2020) of laboratory-made specimens to study deep learning models for use in concrete bridge deck evaluation. The IE data were categorized into three classes: Sound (S), Defective (D), and De-bonded (DB). There were 736 D classes, 715 DB classes, and 2092 S class samples. IE2020 was an effective dataset for deep learning model development; however, the defects were artificial, which could negatively affect a model's accuracy if used to classify impact echo (IE) data from real bridges [1,2]. Kalogeropoulos et al. (2013) collected GPR data for 0.08 m thick concrete slabs exposed to chloride contamination. Cores were taken from the concrete slabs for validation using a drilling rig, and free chloride ion content was calculated using the water extraction procedure for eight slices of 0.01 m each [8]. It is noteworthy that traditional methods such as destructive testing alters the structural condition of the deck and cannot provide assessment information over the entire deck. Dinh et al. (2016) used GPR data collected from twenty-four in-service bridge decks during the Federal Highway Administration's (FHWA's) Long-Term Bridge Performance (LTBP) Program. The data were collected with a ground-coupled 1.5-GHz GPR antenna on cast-in-place concrete bridge decks. The study's objective was to characterize a corrosive environment and provide an overall bridge deck condition assessment; however, this dataset was validated with other NDE methods without developing ground truth for the actual bridge condition [9]. Liu et al. (2020) collected GPR data and converted them into segmented grayscale images sized at 300 × 300 pixels. The final dataset contained 3992 images of 13,026 rebar targets, of which 2370 images were utilized for training and the others for testing. The dataset had two categories, hyperbola, and background, and was labeled using the bounding box method [10]. Mei et al. (2020) claimed that using a bounding box is not acceptable for annotating defects due to irregular crack shapes since too many details are lost if a rectangular bounding box is used to depict cracks. The few available open subsurface defect datasets are predominantly from laboratory specimen data [7].

Publicly available datasets designed to evaluate crack and delamination detection algorithms are limited. Most of these datasets have been processed and simplified since they do not depict real-life scenarios [11]. They manually exclude disturbances and focus only on pavement surfaces using static images [12–14], and others are not publicly available or validated with ground truth [12,15]. A summary of existing structural defect datasets and their descriptions are listed in Table 1. Few open-source datasets contain visual images for crack detection with few pixel-level annotations [12–15].

To the best of the authors' knowledge, there are no publicly available annotated NDE datasets based on different levels of delamination in reinforced concrete bridge decks. They are also not validated by the actual state of delamination in the field. Therefore, Structural Defect Net (SDNET) 2021 is developed and presented. This dataset was collected in realistic conditions to represent the challenges faced by bridge inspectors, such as change in weather conditions, significant environmental effects, and noise, including shadows, occlusion, stains, texture difference, and low contrast due to overexposure; blurring effects due to unmanned aerial system (UAS) motion and poor lighting conditions; and disturbance inclusion during data collection.

**Table 1.** Available open-source datasets.

| Data Type and Description | Defect Types | Material or Structure | Annotation Method | Limitation | References |
|---|---|---|---|---|---|
| | | RGB Images (Surface Defects) | | | |
| Image-56,000 sub-images (256 × 256 px) | Crack (widths from 0.06 to 25 mm) | Concrete bridge decks, walls, and pavements | Labeling | Limited to crack defects only. Not validated with ground truth. | [3,4] |
| 40,000 images with 227 × 227 pixels generated by a 4032 × 3024 resolution camera | Cracks on buildings | METU campus buildings | Labeling | Dataset is based on buildings only. | [16] |
| CFD contains 118 RGB and AigleRN database contains 38 gray-level images. | Cracks | Asphalt pavements | Labeling | Only surface defects. | CrackForest Dataset and AigleRN [17] |
| 600 RGB images | Cracks | Pavement | Pixel level annotation | Only surface defects. | EdmCrack600 [7] |
| At least 17,754 RGB images | Cracks, spall, exposed bars, corrosion stain | Concrete bridges | Bounding box labeling | Only surface defects. | COncrete DEfect BRidge IMage (CODEBRIM) dataset [18] |
| 6500 3D pavement images | Cracks | Asphalt pavement | Labeling | Not publicly available and limited to asphalt pavement. | [19] |
| 7237 RGB images of pavement sections extracted from Google Map | Structural cracks | Asphalt pavement surface | Bounding box labeling | Not publicly available. Dataset not validated with ground truth. Without delamination defects. | [20] |
| | | NDE (IRT/IE/GPR) Subsurface Defects | | | |
| Impact Echo- 2016 IE signals. | Debonding and subsurface defects | Laboratory concrete specimens | Signal labeling | Limited to laboratory specimens. Dataset not validated with ground truth. | [1,2] |
| GPR signals | Chloride migration detection | Laboratory concrete decks | Signal labeling | Limited to laboratory specimens and validated with a destructive method (core samples). | [8] |
| GPR signals collected during the FHWA's LTBP Program | Characterize the corrosive environment | Asphalt and concrete bridge decks | Signal Labeling | Dataset was not validated with ground truth but was validated with other NDE methods and bridge decks. | [21] |
| GPR signals converted into 3992 grayscale images | Rebar Detection and localization | Residential buildings under construction | Bounding box labelling | Dataset not validated with ground truth. | [10] |
| 500 infrared images. | Sub-surface delamination | Reinforced concrete bridges | Semantic pixel-wise image labeling | Dataset is not publicly available. and was not validated. | [5] |

SDNET2021 contains three NDE data types (IRT, IE, GPR) for five existing in-service bridges with validated ground truth. This publicly available rare and state-of-the-art validated dataset will be essential for AI model benchmarking, development, training, and testing needed to evaluate, monitor, and assess bridge conditions.

This paper presents the research activities performed by the authors to achieve this goal. The rest of the article is organized into the following sections: Experimental program, ground truth, quality assessment of the dataset, results, discussion, and conclusion.

## 2. Experimental Program

The North Dakota Department of Transportation (NDDOT) scheduled deck repairs of five 47–49-year-old bridges during the Summer of 2020. These bridges were built to carry I-29 traffic, except for the Park River median (PR MD), designed to provide access to the local rest area (Figure 1). The bridge lengths ranged from 64 m for the Forest River North and South Bounds (FR NB and FR SB) bridges to 142 m for the Park River North and South Bounds (PR NB and PR SB) bridges. A summary of the bridge information is listed in Table 2. The concrete bridge decks are supported on steel beam girders with expansion joints at appropriate intervals. The bridges were inspected via chain-dragging in summer of 2015, 2019, and 2020. This investigation only focused on the bridge decks and did not include the steel girders, vertical supporting piers, or the sub-structures. In addition, the concrete bridge deck had no asphalt overlay at the time of the investigation.

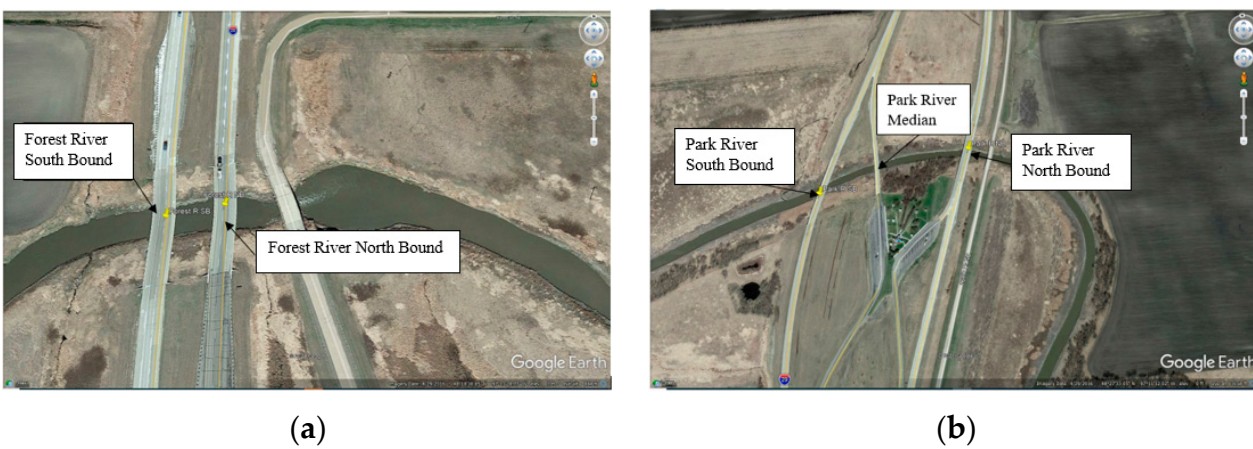

(**a**)                                                    (**b**)

**Figure 1.** Layout map depicting the (**a**) Forest River bridges and (**b**) Park River bridges. (Source: Google Maps).

**Table 2.** Summary of bridge condition data collected by NDDOT.

| Bridge ID | Structure Number (Year Built) | Width (m) × Length (m) | Deck Area (sqm) | Delamination % |
|---|---|---|---|---|
| FR SB | 0029168629 L (1971) | 12.7 × 64 | 816 | 18% 40.4% |
| FR NB | 0029168632 R (1971) | 12.7 × 64 | 816 | 23% 31.5% |
| PR NB | 0029179087 L (1973) | 11.3 × 141.7 | 1806 | 24.56% 26.1% |
| PR M | 0029179123 M (1973) | 7.3 × 111.3 | 977 | 21.0% 34.5% |
| PR SB | 0029179147 R (1973) | 14.9 × 120.4 | 1974 | 3.5% 30% |

Note—Forest River South Bound (FR SB), Forest River North Bound (FR NB), Park River North Bound (PR NB), Park River South Bound (PR SB), Park River Median (PR M), L (Left), R (Right), M (Median).

### 2.1. NDE Data Collection

### 2.1.1. IRT Data Collection

Infrared thermography (IRT) is a noncontact technique to detect subsurface delamination. IRT cameras convert electromagnetic radiation emitted from a region to temperature. The rate at which this energy is emitted is a function of the material's temperature and emissivity. A material's emissivity defines the correlation between the actual kinetic temperature and the object's radiant temperature, which can be changed if the object has delamination. Equation (1) shows the total energy emitted by an object [22].

$$M = \varepsilon \, \sigma \, T^4 \tag{1}$$

where $M$ = Total energy emitted from the surface of a material, $\varepsilon$ = Emissivity, $\sigma$ = Stefan-Boltzmann constant, and $T$ = Temperature of the emitting material in Kelvin. In IRT images, defected areas will emit electromagnetic radiation at a different rate than the intact areas which are manifested in their pixel intensities. Thermography analysis operates under the first law of thermodynamics, which states that all objects with a temperature higher than absolute zero emit radiation in the infrared wavelength range of 700 nm–1 mm, which corresponds to frequencies of 430 THz–300 GHz, between the visible radiation and microwave ranges.

IRT data were collected using a UAS mounted with a thermal camera at an average altitude of 18 m above ground level. Figure 2 shows IRT data collection using a UAS and collected sample image. Table 3 shows the weather conditions and the hours of solar exposure from the time of sunrise during IRT data collection. The specifications of the IRT sensor are shown in Table 4. 1064 IRT images with a resolution of 640 × 512 pixels were collected from the in-service bridge decks.

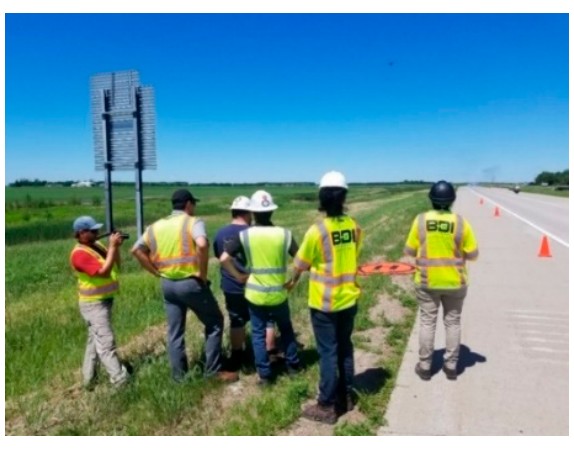
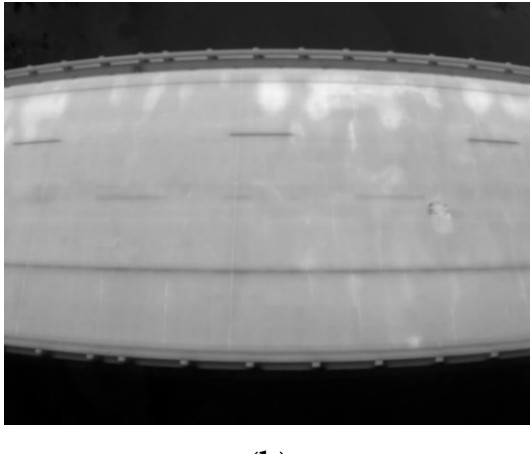

(**a**)          (**b**)

**Figure 2.** IRT data collection depicting (**a**) crew collecting data with a UAS and (**b**) sample of IRT image.

**Table 3.** Ambient weather condition and inspection dates for IRT data collection.

| Bridge ID | Time | Temperature (°C) | Sun Light Exposure (Hours) | Humidity (%) | Wind Speed (kmph) |
|---|---|---|---|---|---|
| FR SB | 9:55–10:25 a.m. | 26.0 | 3.0 | 47.0 | 10.5 |
| FR NB | 10:26–10:44 a.m. | 26.7 | 3.5 | 44.0 | 12.9 |
| PR NB | 11:36–11:55 a.m. | 27.0 | 4.5 | 47.0 | 12.9 |
| PR MD | 12:09–12:32 p.m. | 27.8 | 5.0 | 44.5 | 14.5 |
| PR SB | 12:34–12:55 p.m. | 27.8 | 5.5 | 45.0 | 16.1 |

**Table 4.** UAS and IR camera specifications.

| Characteristics | Specifications |
|---|---|
| Thermal Resolution: | 640 × 512 pixels |
| Full Frame Rates: | 30 Hz (NTSC) 25 Hz (PAL). |
| Spectral Band: | 7.5–13.5 μm. |
| Pixel Pitch: | 17 μm. |
| Thermal Imager/Detector type: | Uncooled VOx Microbolometer. |
| Digital Zoom | 2×, 4× |
| Field of View | 24° × 19° |

### 2.1.2. IE Data Collection

IE is a nondestructive testing (NDT) method commonly used for evaluating concrete structures [1,2]. IE technique uses elastic waves to identify and characterize delamination in concrete structures using the transient vibration response of a plate-like structure subjected to a mechanical impact.

The transient time response of a solid structure is measured with an accelerometer placed on the surface close to an impact source. IE was implemented for deck evaluation by conducting point testing on a pre-defined grid. The set-up of the IE equipment used comprises (1) NI-USB-4431 USB DAQ System. (2) Laptop w/LabView software. (3) USB-A to USB-B Cable. (4) Accelerometer w/mounting base. (5) Accelerometer BNC Cable. (6) Impact Hammer (Figure 3a(1–6)). The test was carried out on 0.3 m × 0.3 m spacing grids. Due to the limited time for data collection before the decks underwent repairs, IE data were collected from all three 3 m by 3 m regions on each deck. Therefore, an overall 363 points for three areas were tested for the FR SB bridge. A similar number of points were also tested for the region of interest of other bridge decks.

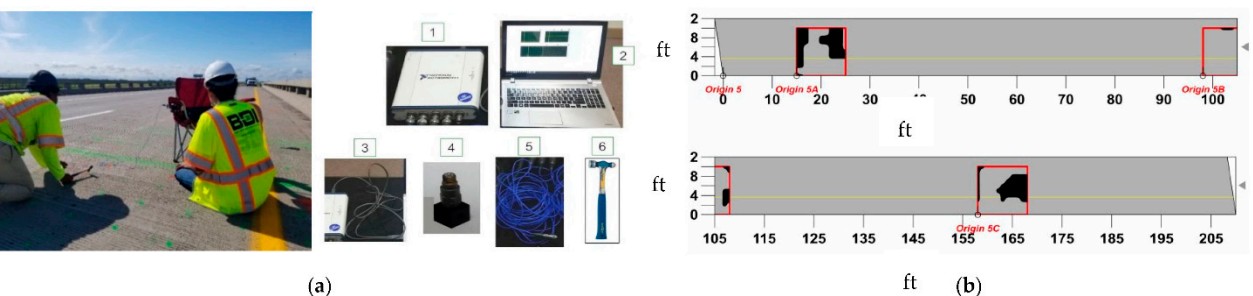

(**a**)  (**b**)

**Figure 3.** IE data collection depicting (**a**) crew and set up for collecting data (**b**) IE data collection layout for FR SB.

The output of the IE signals is time and acceleration (g) with 204,800 rows, each testing point saved in an 'lvm' file format. Overall, 1936 IE test points (files) were taken for the region of interest evaluated for the bridge deck sections. Also, the time duration varied from a minimum value of zero (0) to a maximum of 1.99990, having a time step of 9.765625E-6 s (Approximately 0.00001 s). Figure 3a,b shows IE set-up, data collection, and layout for three regions of tests.

### 2.1.3. GPR Data Collection

GPR is a commonly used NDE technique for locating reinforcement bars, corroded bars, and corrosion-induced delamination in concrete structures [10,21]. The GPR method transmits electromagnetic radio waves, with frequencies ranging from a few MHz to a few GHz, then records the reflections to determine dielectric constants of a medium, such as a rebar or delamination, which should be different from concrete. The GPR data consist of changes in reflection strength and the arrival time of specific reflections, source wave distortion, and signal attenuation [21]. The data were collected with the GSSI GPR Equipment set-up comprising: (i) SIR-3000, (ii) 2600 MHz antenna, (iii) 11-pin black

cable, (iv) 19-pin blue cable, (v) Lithium-Ion Battery, (vi) Battery charger (Figure 4a(i–vi)). The number of transverse and longitudinal scans for the FR NB bridge were 6 and 22, respectively. The transverse scans were conducted at a 0.6 m interval, while the longitudinal scans were conducted at a 3 m interval. The spatial resolution was proposed by the GPR operators to ensure maximum data collection. Figure 4a,b shows the equipment and sample scan layout. Overall, 209 scans were conducted for the entire bridge decks.

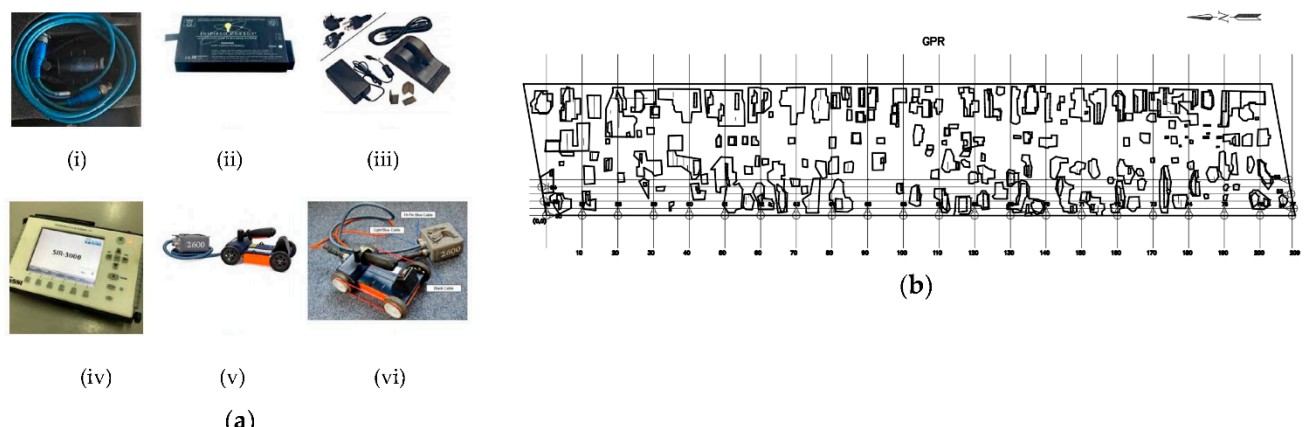

**Figure 4.** GPR data collection depicting (**a**(**i**)–(**vi**)) scan for GPR data collection (**b**) GPR scan lines.

The GPR signals have a vertical time scale of 12 ns and 512 samples per scan. The longitudinal scans along the length of the bridges gave a maximum output file having 16,383 amplitudes. The transverse signals along the width of the bridges gave an output file having about 1225 amplitudes.

The environmental condition during IE and GPR data collection is shown in Table 5. The average temperature and humidity during IE and GPR data collection are shown in Table 6.

**Table 5.** IE and GPR data collection ambient weather and deck condition.

| Bridge ID | Date Collected | Temperature (°C) | Humidity (%) | Deck Condition |
|---|---|---|---|---|
| FR SB | 6 July 2020 | 18.3 | 67.0 | Dry |
| FR NB | 7 July 2020 | 27.8 | 43.0 | Dry |
| PR NB | 8 July 2020 | 25.6 | 66.5 | Dry |
| PR M | 7 July 2020 | 23.1 | 55.0 | Dry |
| PR SB | 9 July 2020 | 22.8 | 56.0 | Dry |

**Table 6.** Summary of NDE data collected for bridges.

| Data Collection | Data Types and Formats | FR-NB | FR-SB | PR-NB | PR-SB | PR-MD | Number of Files |
|---|---|---|---|---|---|---|---|
| Images (Round 1) | Thermal Image (JPEG): | 122 | 66 | 76 | 95 | 121 | 480 |
| Images (Round 2) | Thermal Image (JPEG): | 76 | 84 | 48 | 152 | 100 | 460 |
| Images (Round 3) | Thermal Image (JPEG): | 19 | 16 | 24 | 31 | 34 | 124 |
| GPR | Downloaded (csv,DZT,DZX) | 28 | 29 | 50 | 53 | 49 | 209 |
| IE | Downloaded (lvm) | 415 | 415 | 440 | 542 | 466 | 2275 |

## 2.2. Ground Truth

The inspection crew surveyed all five bridge decks to identify the subsurface delamination locations and sizes using chain dragging. The concrete toppings of all five bridge decks were removed by milling the top 75 mm of each deck prior to chain dragging. The crew chain-dragged on the deck, and the suspected delaminated locations were marked and mapped using GPS (Figure 5a). The portions without delamination were classified as

sound (class 1). The marked regions were then exposed just above the top reinforcement bars using a jackhammer. These patches and portions were classified as class 2 removal (Figure 5b). These regions were then chain dragged to detect possible deeper delamination and then further exposed below the rebar reinforcement. This deeper exposure is called class 3 removal (Figure 5c). The delamination class definitions used to annotate the collected data are class 1 Sound (No Delamination), class 2 Delamination (delamination above top bar mat), and class 3 Delamination (delamination below top bar mat). The typical bridge layout section showing the class definition for delamination removal is illustrated in Figure 6.

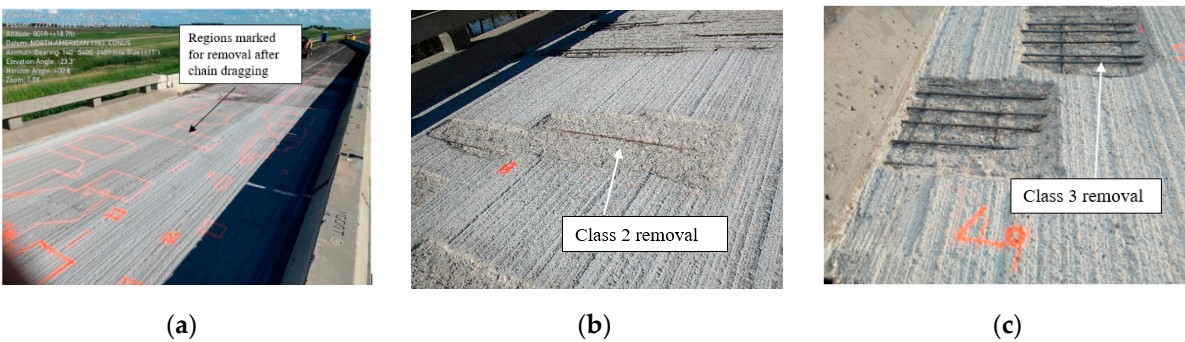

| (**a**) | (**b**) | (**c**) |

**Figure 5.** (**a**) Delaminated section markings for FR bridge deck (**b**) class 2 sub-surface delamination removal and (**c**) class 3 sub-surface delamination removal.

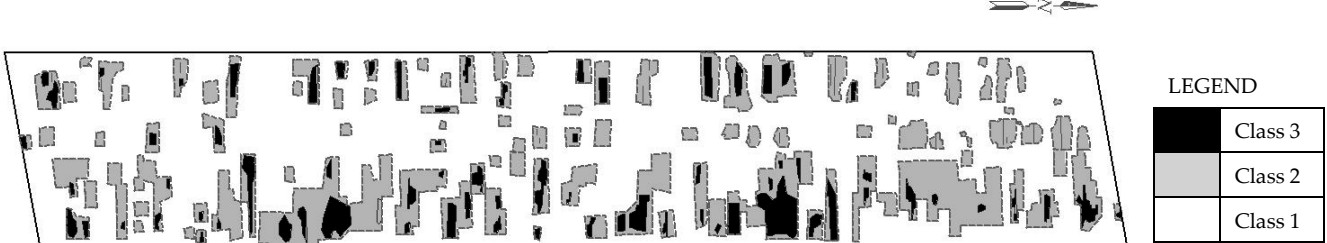

**Figure 6.** Layout plan showing delamination survey for the FR SB bridge.

A set of delamination maps were generated for each bridge deck, indicating the delamination locations and classes. These maps were used as the ground truth for the NDE data classification. Classes 2 and 3 are distinguished from sound concrete by dashed and solid boundaries, respectively.

### 2.3. NDE Data Acquisition Conditions

NDE data collection needs to be conducted in a manner to maximize their efficiency for deck evaluation. In this section, the authors have reviewed data collection standards and past publications, summarized in Table 7, to ensure the NDE data were collected according to standards, specifications, and established methodologies.

Reliable NDE data collection depends on favorable environmental, deck surface conditions, sensor specifications, altitude, and other related conditions (Table 7). The data collection for this study has been carried out considering these factors and conditions. Data were collected based on reviews of past studies and standard specifications, which are relatively consistent with the existing practice.

**Table 7.** Comparison of parameters for NDE data collected in this study with past studies.

| NDE Type | Condition Type | | Refences |
|---|---|---|---|
| IRT | Temperature | 26–27.8 °C | [23,24] |
| | Solar exposure | Minimum of 6 h | |
| | Wind speed | 11–16 km/h | |
| | Ambient temperature | No testing when tempt is less than 0 °C | |
| | UAS AGL | 15–18 m AGL | |
| | Image overlap | 65–80% | |
| IE | Temperature | 18.3–27.8 °C | [25,26] |
| | Deck condition | Concrete surface-dried and cleared of debris. | |
| | Grid size | 0.3 m × 0.3 m test grid | |
| | Contact time | | |
| GPR | Temperature | 18.3–27.8 °C | [27,28] |
| | Deck condition | Concrete surface-dried and cleared of debris. | |
| | | 2600 MHz. Equipment: | |
| | Antenna | GSSI SIR-3000 Data Acquisition System. | [21] |
| | | GPR Antenna | |

### 2.4. Quality Assessment of Dataset

The performance of an AI model constructed using a dataset is often influenced by the data's quality; therefore, a set of generic quality assessment metrics were used to assess the quality of the IRT, IE, and GPR data acquired in this study. The data collected are classified into time series and image data; therefore, appropriate quality control methods were adapted and used for each data type.

#### 2.4.1. Signals

Sample plots of amplitudes for the IE and GPR signals are shown in Figure 7a–d.

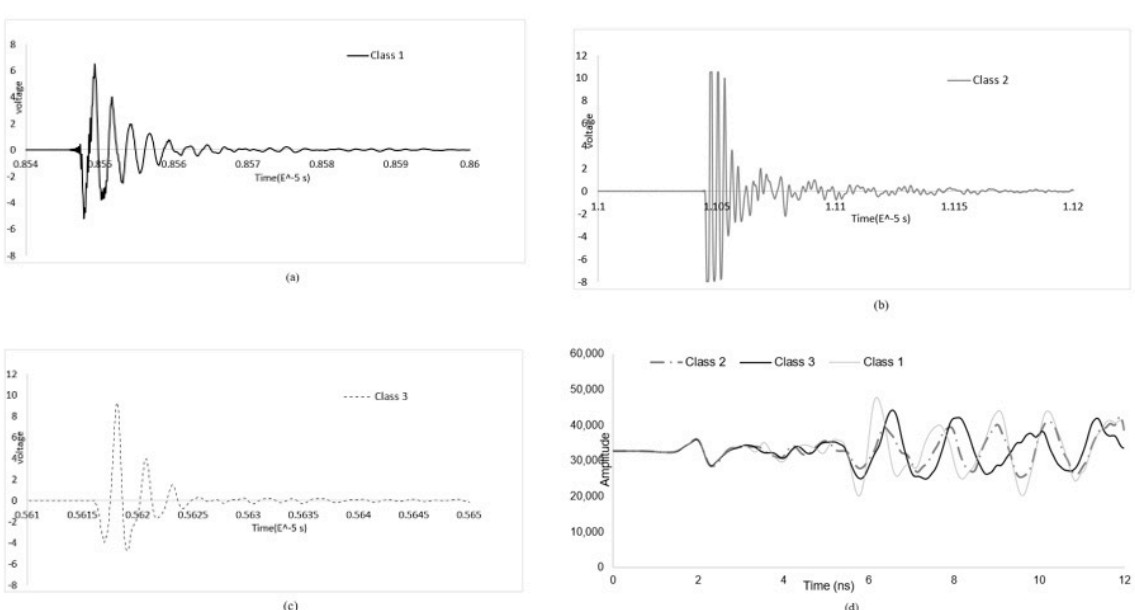

**Figure 7.** Selected samples of (**a**–**c**) IE signal for PR M classes 1, 2 and 3, and (**d**) GPR signal for PR file-004 for FR SB.

The IE signal outputs are time and acceleration (g) with 204,800 rows and each testing point saved in a .lvm file format, totaling 1936 IE test points (files) for all bridge sections. The IE data were checked for null, duplicate, void/missing values, correctness, repetitiveness, and other pre-processing operations deemed fit for quality checks [29]. The same quality checks were performed on the GPR raw data. Test profiles were inspected in their raw format for data quality to ensure no trace data gaps were present. The correlation coefficient matrix for the raw GPR signals was computed. The investigation showed similarity and

a high correlation coefficient of at least 0.8 between signals of the same class and a lower coefficient of about 0.5 between signals of different classes. Figure 7a–d depicts plots of some of the signals generated from tests collected from regions with different degrees of subsurface delamination.

### 2.4.2. IRT Quality Assessment

Subjective and objective methods are the two broad classifications for image quality assessment. A non-reference objective assessment metric was needed in this study since subjective methods are based on individual opinions. Common non-reference quality assessment metrics for images are Perception Based image quality evaluator (Pique), Naturalness image quality evaluator (Niqe), and Blindness/referenceless image spatial quality evaluator (Brisque). The metrics are used to evaluate raw IRT image quality in this study. These metrics compare an image or set of images to a default model computed from images of natural scenes with similar distortions. A smaller Piqe, Niqe, and Brisque score indicates better perceptual quality [30,31]. Niqe and Brisque are based on spatial features derived from natural scene statistics. Niqe compares an image to a default model computed from images of natural scenes. Figure 8a,b show sample IRT images collected at different ambient weather conditions and quality categories of excellent and good Brisque scores. Figure 8c,d depict sample IRT images with fair and poor Pique scores, Figure 8a show thermal image samples having poorly natural/unnaturalness, and Figure 8b show thermal image samples having excellent/good naturalness.

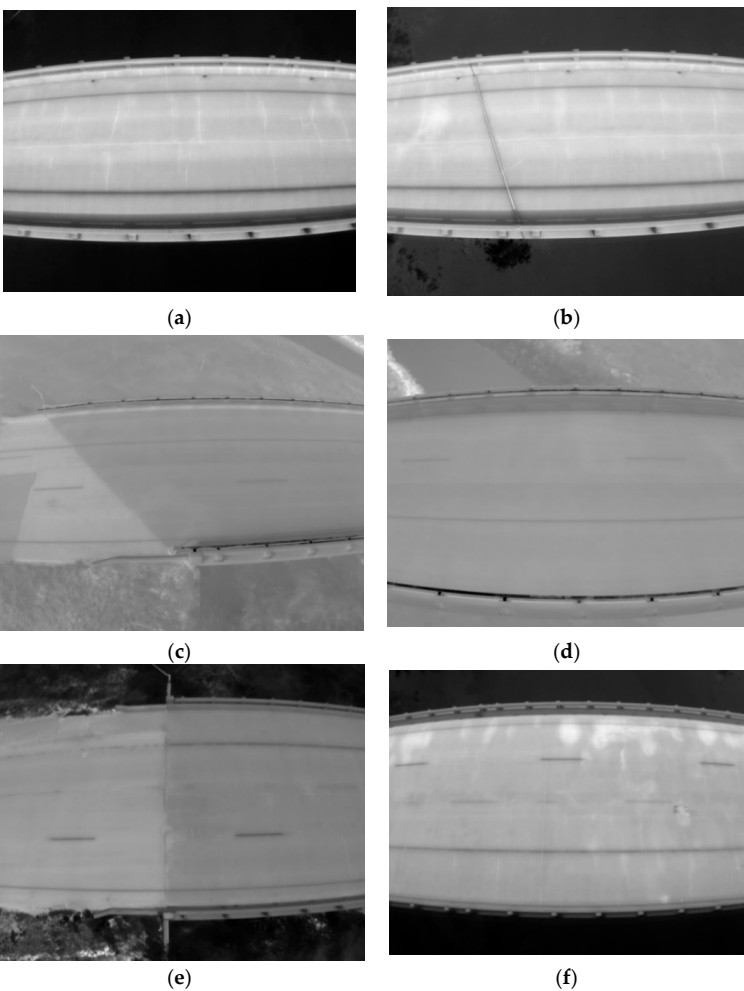

**Figure 8.** Samples of IRT images with (**a**) Excellent and (**b**) Good Brisque score; (**c**) Fair and (**d**) Poor Pique Scores (**e**) Poor natural or unnaturalness and (**f**) Excellent or good naturalness.

The quality check results for the collected IRT, IE and GPR data are summarized in Table 8. The IRT data indicates that Piqe, Niqe and Brisque quality metrics are satisfactory with excellent/good values ranging from 82–100%.

**Table 8.** NDE data quality assessment.

| Metrics % | Excellent/Good % | Fair % | Poor/Bad |
|---|---|---|---|
| Image Data Quality | | | |
| Piqe | 97 | 3 | 0 |
| Niqe | 82 | 18 | 0 |
| Brisque | 100 | 0 | 0 |
| **GPR and IE signals** | | | |
| Null Values | 100 | 0 | 0 |
| Missing values | 100 | 0 | 0 |
| Duplicate values | 100 | 0 | 0 |

## 3. Results and Discussion

### 3.1. Data Annotation

Data annotation is the process of assigning the collected NDE data to one of the defined delamination classes using the ground truth maps from Section 3. This dataset's annotation was created with Autodesk Civil 3D, Agisoft photo scan, and a computer program developed in the MATLAB R2020b software package.

### 3.2. IRT Image Annotation

Image annotation can be performed in three different ways: image labeling, where an entire image is labeled with a particular class; object detection using bounding boxes, where a rectangular box is placed around a group of pixels in each class; and semantic segmentation, where each pixel is assigned to a particular class. Semantic segmentation for defects detection purposes provides the most information about the data; however, it is the most time-consuming since every pixel must be labeled [32]. The main challenge for semantic segmentation of IRT images is to stitch individual images, so they match each deck's defect map depicted in Figure 6. Therefore, we developed a pixel-based semantic annotation method to annotate IRT images autonomously. The main idea was to superimpose the ground truth maps to thermal stitched maps for each bridge. Fast and accurate image annotation in a semantic manner remains an open problem in computer vision and related fields; however, the procedure developed in this study can be effectively used to assign delamination class to each pixel accurately.

Other semantically segmented image datasets [33,34] rely on image labelers to assign labels to pixels. These methods can be time-consuming and labor-intensive proportional to the level of detail required and possible inconsistencies between different annotators. The annotation adopted in this study can effectively remove the role of the IRT image labeler.

The major steps devised for IRT image annotation are presented in Figures 9 and 10a–g. The individual images are stitched together to generate a single-view presentation of the thermal images for the entire deck. The authors used commercial software (Agisoft-metaphase 2021 © professional-student trial version, Agisoft LLC, St. Petersburg, Russia, 191015) to properly create stitched maps for each bridge deck (Figure 10a,b). The authors generated the stitched map by adopting relevant metadata for the set of selected images that produced the highest quality. This process can also be completed using computer vision techniques to remove lens distortion, extract features, and stitch the images together. The generated stitched maps for each bridge were aligned with their corresponding ground

truth maps (Figure 10c,d), which required the use of the geometrical transformations shown in Equations (2)–(4).

$$T = \begin{bmatrix} 1 & 0 & 0 \\ 0 & 1 & 0 \\ X & Y & 1 \end{bmatrix} \tag{2}$$

$$R = \begin{bmatrix} cosd & sind & 0 \\ -sind & cosd & 0 \\ 0 & 0 & 1 \end{bmatrix} \tag{3}$$

$$S = \begin{bmatrix} a & 0 & 0 \\ 0 & b & 0 \\ 0 & 0 & 1 \end{bmatrix} \tag{4}$$

where $T$ is Translation, $R$ is Rotation, and $S$ is scale. Equations (2)–(4) demonstrate the Affine transformation matrix for the translation, rotation, and scale used. 'X' and 'Y' are displacements along the x and *y*-axis, 'd' is the angle of rotation, and 'a' and 'b' are scale factors along the x and *y*-axis which are the same here. Table 9 summarizes the affine transformation tuned parameters adopted prior to image registration.

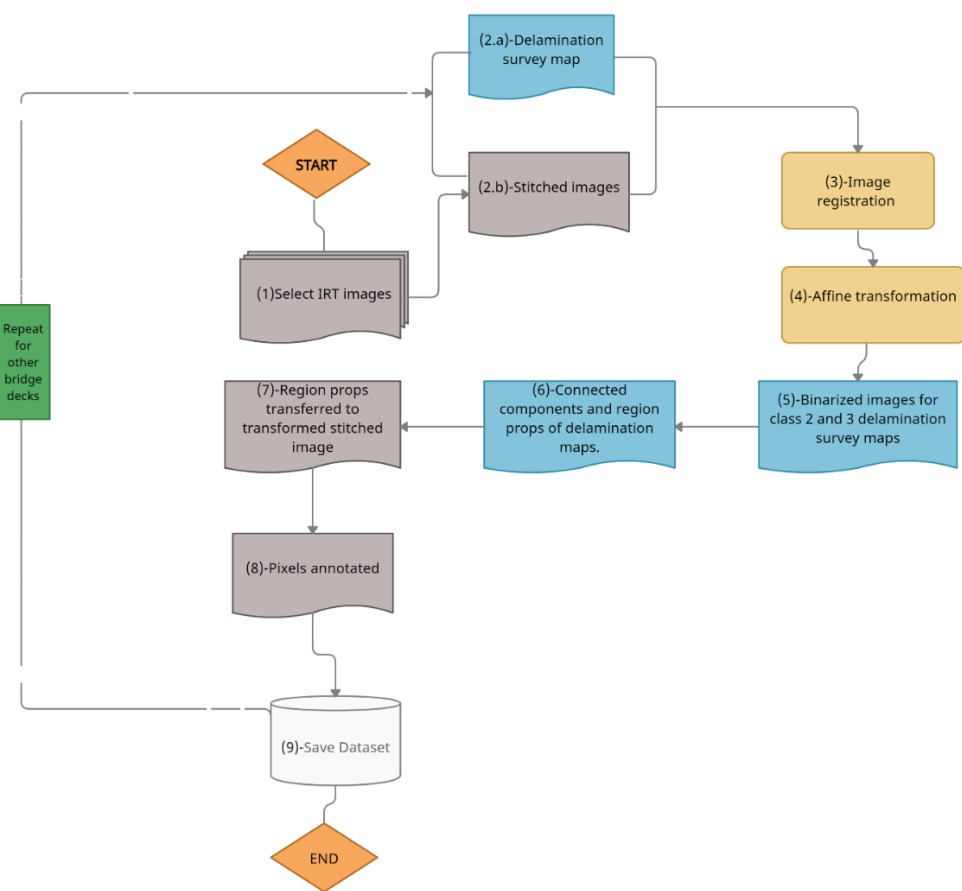

**Figure 9.** Flowchart for image processing and annotation of IRT.

The authors developed an algorithm to register and transform the geometric properties of the thermal stitched maps on the ground truth images. The original defect maps were in CAD format. However, they were converted to RGB images for computer vision processing.

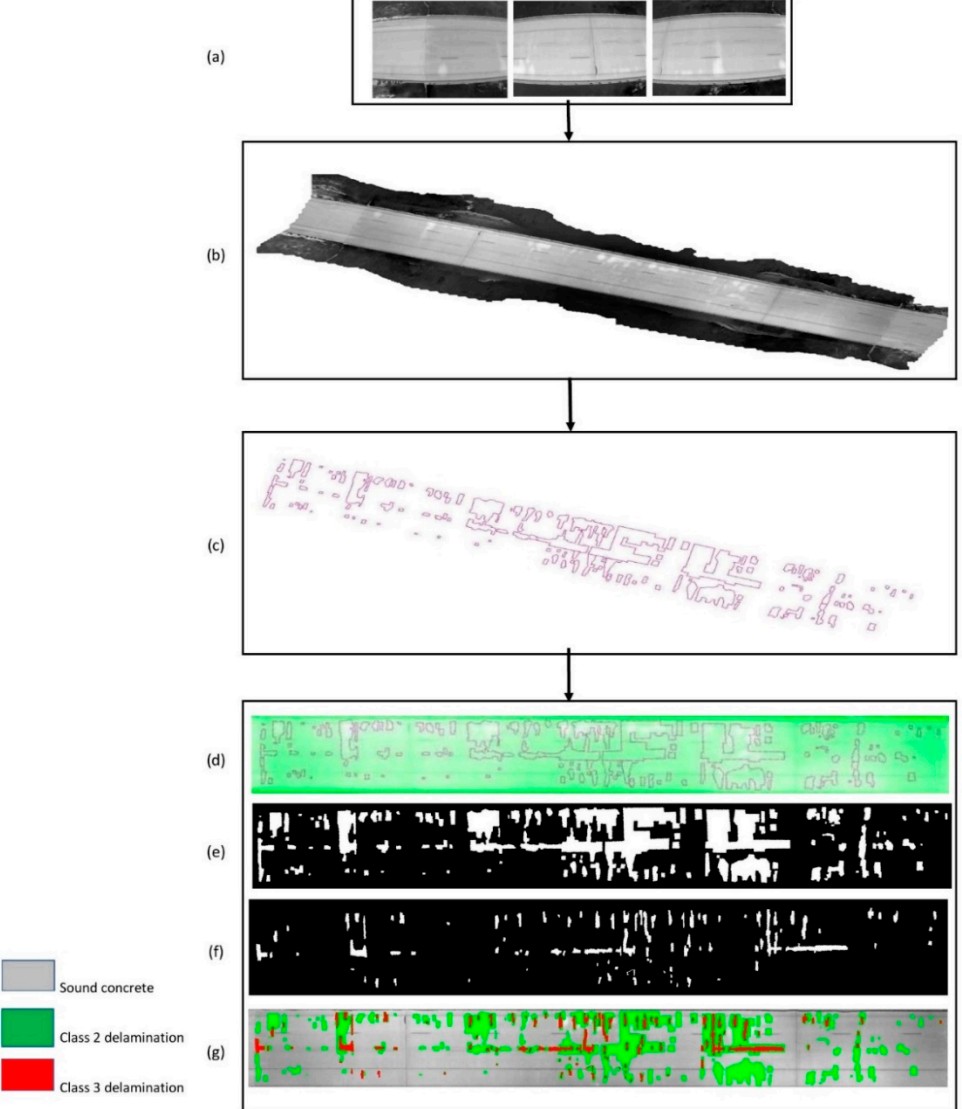

**Figure 10.** Workflow for annotation (**a**) raw images inputs for stitching (**b**) stitched image (**c**) ground truth survey map for class 2 (**d**) aligned and registered ground truth (**e**,**f**) binarized images for class 2 and 3 delamination, and (**g**) Annotated pixels.

**Table 9.** Summary of Affine transformation tuned parameters for registration.

| Bridge ID | Translation (X, Y) (Pixels) | Rotation (d) (degree) | Scale (a,b) |
|---|---|---|---|
| FR SB | [−474, 220] | −4.5 | 1.33 |
| FR NB | [−200, 105] | −1.9 | 1.12 |
| PR NB | [−105, −5] | −87.4 | 2.6 |
| PR M | [0, 0] | 2.8 | 1 |
| PR SB | [−360, −20] | −2 | 2.88 |

The locations of the pixels within these regions were extracted for each class, resulting in two binary images representing classes 2 and 3 (Figure 10e,f). Since the stitched maps were aligned with the ground truth maps, these pixels show the actual location of class 2 and 3 removals on the stitched maps, as shown in Figure 10e,f. These were later superimposed on the IRT maps, as shown in Figure 10g. The pixels in the final image have been annotated

pixels-wise as G (0 255 0) for all the class 2 delamination pixels, R (255 0 0) for the class 3 delamination pixels, and all others as class 1 or sound. The class 2 and 3 pixels are denoted as green and red, respectively.

### 3.3. IE Annotation

The IE datapoints are point-wise measurements, annotated and validated by cross-referencing the IE signal locations on the ground truth. Figure 11a–d illustrates the IE test regions for the FR NB bridge deck layout with the removal classes. The IE tests were performed in a 3 m × 3 m marked region at 0.3 m intervals on each bridge. Three regions, A, B, and C were defined on the FR NB bridge. The exact locations of the IE test points were mapped on the ground truth layout. Each signal was annotated automatically based on the class within which it was tested (Figure 11e–g). The annotation output was cross-referenced with the ground truth map for consistency.

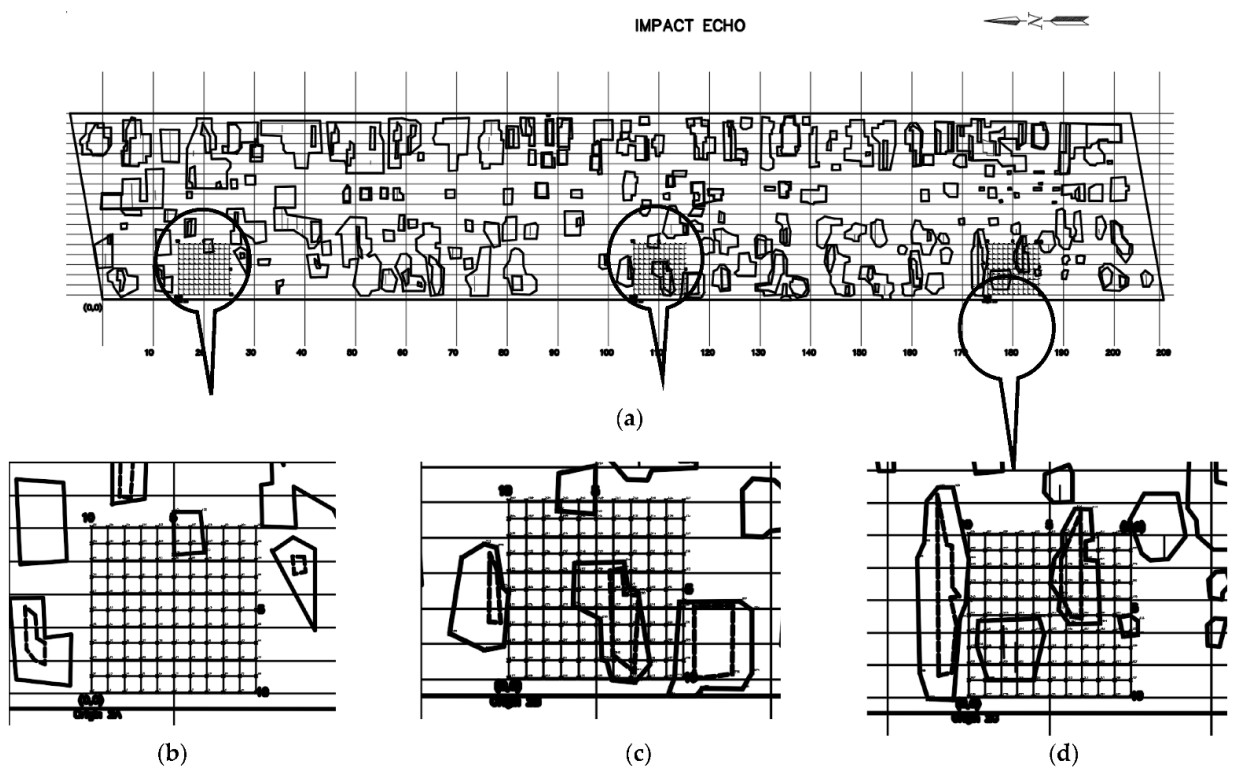

**Figure 11.** (**a**) IE test points (**b**–**d**) Regions A, B and C IE test regions on ground truth.

### 3.4. GPR

We used a similar approach to IE data annotation for the GPR signals; however, the GPR device scanned each bridge deck along a designated scan line, unlike IE. Each GPR signal coordinate was extracted before annotation using Equations (5)–(8).

$$d_{ux} = \frac{L_x}{n} \tag{5}$$

$$d_{uv} = \frac{L_y}{n} \tag{6}$$

$$X_i = X_{x-i} + d_{ux} \tag{7}$$

$$Y_i = Y_{y-i} + d_{uy} \tag{8}$$

where

$d_{ux}$, $d_{uv}$ are the signal discretized sub-divisions for longitudinal and transverse signals respectively,

$L_x$, $L_y$ are the length of the signal scans for longitudinal and transverse signals respectively,

$n$ is the number of signal amplitudes,

$X_{x-i}$, $Y_{y-i}$ are the initial coordinate coordinates of the longitudinal and transverse scans, respectively.

$X_i$, $Y_i$ are the cumulative coordinates of the longitudinal and transverse scans, respectively.

The longitudinal and transverse scan lines were plotted on the ground truth maps at 3.3 m and 0.6 m intervals for the bridge deck, as shown in Figure 12, developed to annotate the GPR data. The red-dashed line in Figure 12 shows the Southwards GPR scan direction at 0.6 m from the bridge edge. The intersections of each GPR line scan with the class 2 and class 3 delamination regions were extracted automatically. This was carried out using civil 3 d Autodesk packages. The coordinates generated for the delamination and the discretized signals were imported into the Matlab program for further analysis. The signal point coordinates were classified into classes 1, 2, or 3 according to the union of intersections with the delaminated regions.

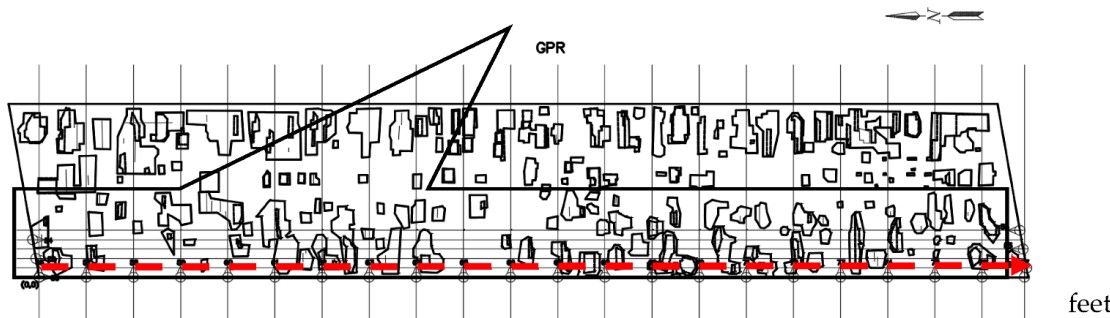

**Figure 12.** GPR scan signal line 51 from CAD layout for FR NB.

The GPR data annotation output was cross-referenced with the mapped regions of the ground truth for consistency, as shown in Figure 12. A summary of the SDNET2021 annotation results is listed in Table 10.

**Table 10.** SDNET2021 Annotation Summary.

| GPR data | | | | | |
|---|---|---|---|---|---|
| classes of delamination | PR M | FR NB | PR NB | PR SB | FR SB | Total signals |
| class 1 | 171,085 | 66,334 | 94,978 | 61,732 | 54,030 | 448,159 |
| class 2 | 56,528 | 39,577 | 26,590 | 38,510 | 29,885 | 177,483 |
| class 3 | 13,478 | 6945 | 443 | 6674 | 7392 | 37,460 |
| Total | 241,091 | 141,500 | 141,500 | 106,916 | 91,307 | **663,102** |
| **IE data** | | | | | |
| classes of delamination | PR M | FR NB | PR NB | PR SB | FR SB | Total signals |
| class 1 | 291 | 301 | 273 | 257 | 257 | 1379 |
| class 2 | 61 | 49 | 74 | 213 | 96 | 493 |
| class 3 | 12 | 13 | 16 | 13 | 10 | 64 |
| Total | 364 | 363 | 363 | 483 | 363 | **1936** |
| **IRT data** | | | | | |
| classes of delamination | PR M | FR NB | PR NB | PR SB | FR SB | Total pixels |
| class 1 | 898,758 | 344,771 | 802,348 | 572,455 | 244,265 | 2,862,597 |
| class 2 | 298,544 | 189,280 | 215,113 | 411,147 | 138,229 | 1,252,313 |
| class 3 | 79,294 | 80,619 | 49,640 | 200,968 | 55,249 | 465,770 |
| Total Pixels | 1,276,596 | 614,670 | 1,067,101 | 1,184,570 | 437,743 | **4,580,680** |

*3.5. SDNET2021 Validation, Processing and Evalsuation*

In this section, the conventional methods for the interpretation and classification of NDE data were applied. The purpose of this dataset is to promote data-driven methods beyond the ones used in this section; however, it is crucial to compare classifications made by the conventional methods with SDNET2021 annotation.

### 3.5.1. IRT Data

While there is no unique method to find delamination in IRT images, irregular temperature differences indicate potential delamination in bridge decks. Delaminated regions emit different thermal energy compared to the sound concrete, which is manifested in terms of change in the pixel intensity. In order to segment the pixels with varying intensities, image processing techniques such as image enhancements, thresholding, binarization, segmentation, etc., can be used. The authors have developed an optimum image processing technique by optimizing the sensitivity parameter for segmentation to produce a binary map [35–37]. Figure 13a,b shows the initial processing of the IRT image for FR SB for delamination detection [35–37]. The preliminary processing results show delamination detection using the developed dataset and benchmarked against the ground truth. However, there are false reports associated with conventional image processing techniques. This investigation also emphasized the need for the development of more advanced AI models that could potentially reach better delamination detection, such as convolutional neural networks.

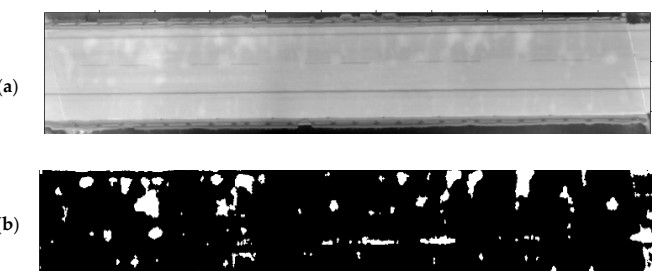

**Figure 13.** IRT Preliminary results (**a**) Original image (**b**) Processed image for delamination detection for FR SB.

### 3.5.2. IE Data

IE signals were preprocessed using the traditional peak frequency and Fast Fourier Transform (FFT) approach as a preliminary processing technique for IE dataset validation in comparison to generated ground truth [38]. The defect maps from the analysis of the raw IE data showed some similarity with the ground truth (Figure 14b). The peak frequencies were used to develop contour maps depicting defective areas. Past studies have revealed that the frequency is proportional to the depth of the concrete deck. Therefore, sound portions will exhibit high-frequency peaks, and delaminated areas will exhibit lower frequency values [1,2]. Figures 14a, 15a, 16a and 17a shows the ground truth delamination map, while Figures 14b, 15b, 16b and 17b depicts the contour maps with different color representations created from the frequency method. For instance, blue coloration signifies regions that have a very high tendency of sound regions, while red signifies higher tendency of defected areas. These detected regions, when compared with the ground truth, showed a relatively accurate consistency in delamination detection, as shown by the bonded boxes for a similar region of interest in Figure 14a,b. However, the difference may be due to several reasons. The IE adopts a pointwise method in detection, contrary to how the ground truth was developed, by mapping the area of the defected regions. Therefore, it makes sense for these two maps to highlight different borders for delamination. Additionally, the peak frequency method is prone to error itself. It has resulted in false delamination detection in the past (Jafari et al., 2021). The comparative analysis in this section shows the importance of having

an annotated IE dataset for more accurate delamination detection. The preliminary IE results indicate that the dataset under development is fit for use.

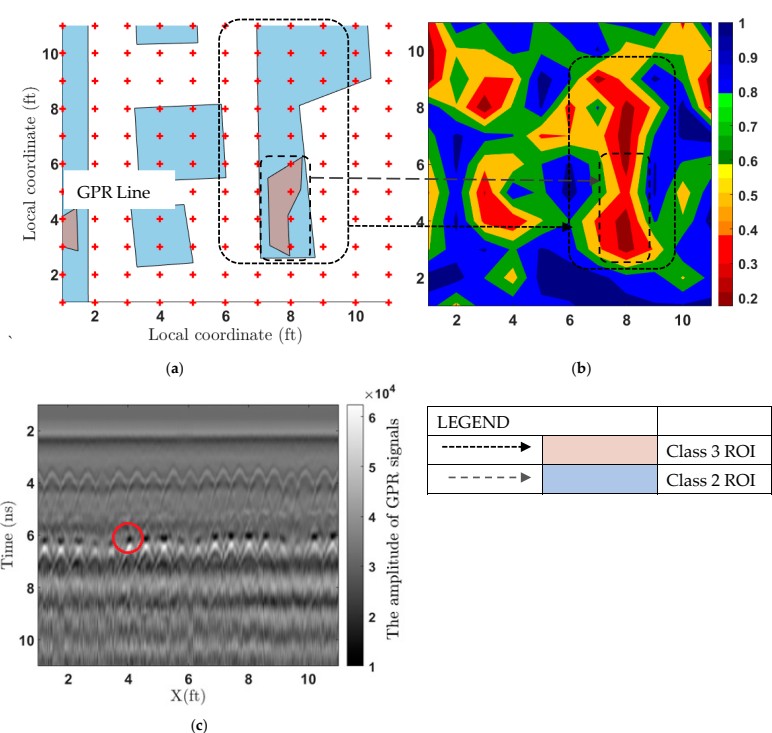

**Figure 14.** FR SB origin A (**a**) ground truth, (**b**) frequency approach, and (**c**) B scan of the GPR profile (local coordinate $Y = 4$).

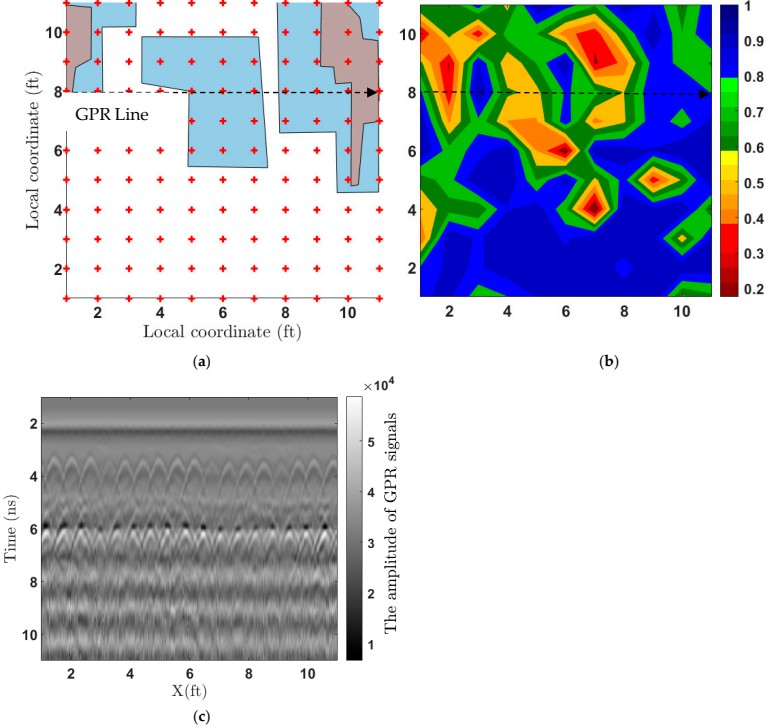

**Figure 15.** FR SB origin C (**a**) ground truth, (**b**) frequency approach, and (**c**) B scan of the GPR profile (local coordinate $Y = 8$).

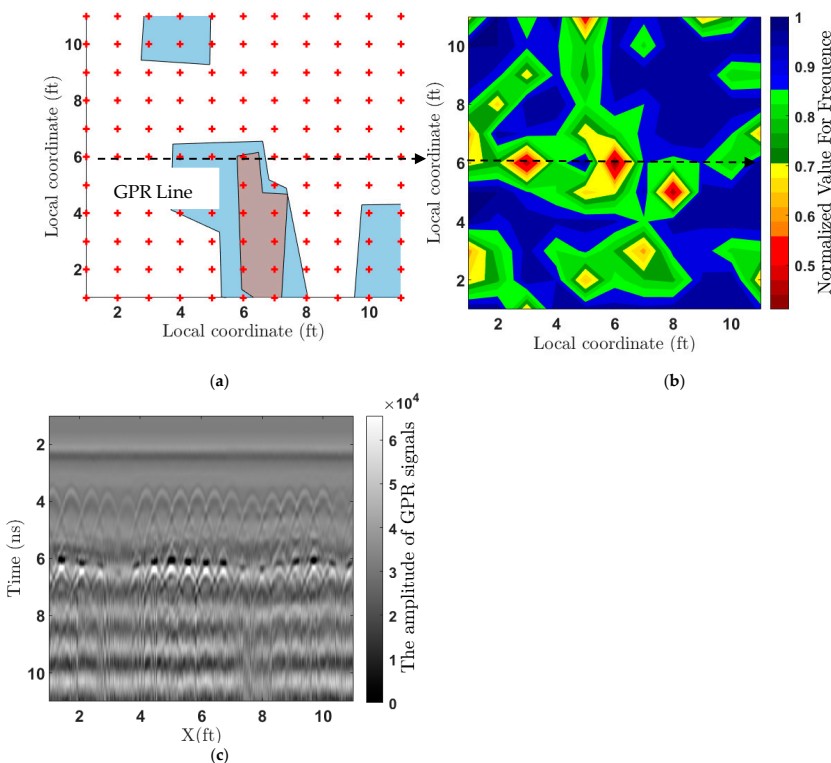

**Figure 16.** FR NB origin D (**a**) ground truth, (**b**) frequency approach, and (**c**) B scan of the GPR profile (local coordinate *Y* = 6).

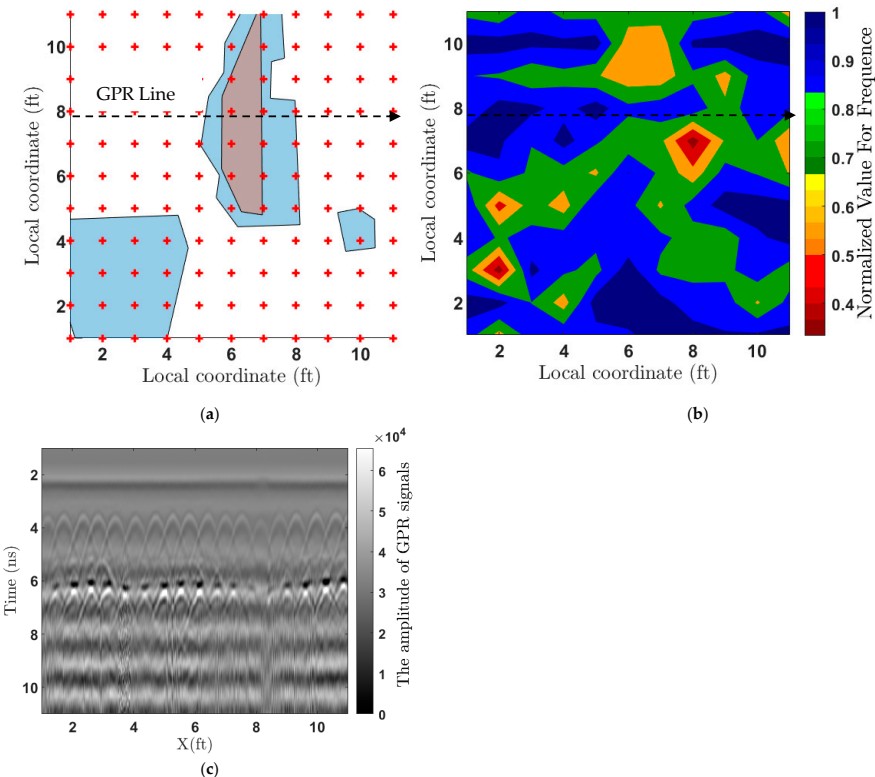

**Figure 17.** FR NB origin B (**a**) ground truth, (**b**) frequency approach, and (**c**), B scan of the GPR Data (local coordinate *Y* = 6).

### 3.5.3. GPR Signal Data

In this study, B scans of GPR signals were developed as one of the conventional methods of GPR analysis [5,21]. In this method, the electromagnetic signal generated from scanning was used to identify the defected regions in the bridge deck. Different materials have been shown to have different dielectric properties. Therefore, scanning along the bridge deck presents the B-scan signals for delaminated and sound portions of the deck. The amplitudes of these signals are converted to 2D images for the concerned ROI and compared with the ground truth, as shown in Figures 14c, 15c, 16c and 17c. The distinguished regions are more visible in the profile when the GPR EM waves reflect off of the defective regions, as marked with a circle in Figure 14c. This may have been caused by the deck's possible voids and material variations [21]. These preliminary results show the possibility of defect detection using the collected GPR dataset. The rebar interface with the concrete is shown by the red circle in Figure 14c.

In addition, the GPR data were validated by plotting the signals from the portions of the deck showing rebar corrosion and section loss with the signals for the sound portions of the deck. Strong rebar reflection represents sound concrete, while weak rebar reflection indicates deterioration [21]. Figure 18 shows three (3) signals each of class 1 (1A, 1B, and 1C-sound concrete), class 2 (2A, 2B, and 2C), and class 3 (3A, 3B, and 3C-sound concrete), delaminated and corroded rebar regions, while Figure 19 shows the corresponding images of the exposed regions during inspection used for validating GPR signals.

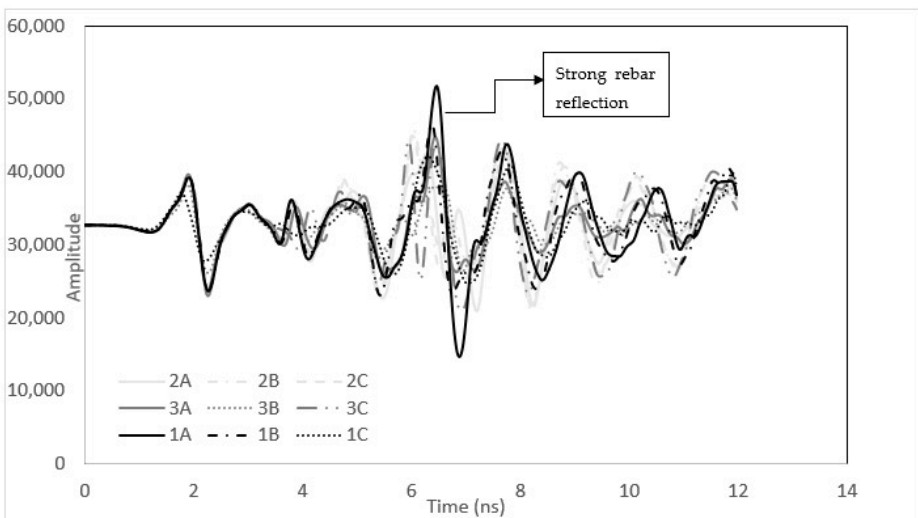

**Figure 18.** Plot showing signals for sound portions and corroded rebar regions.

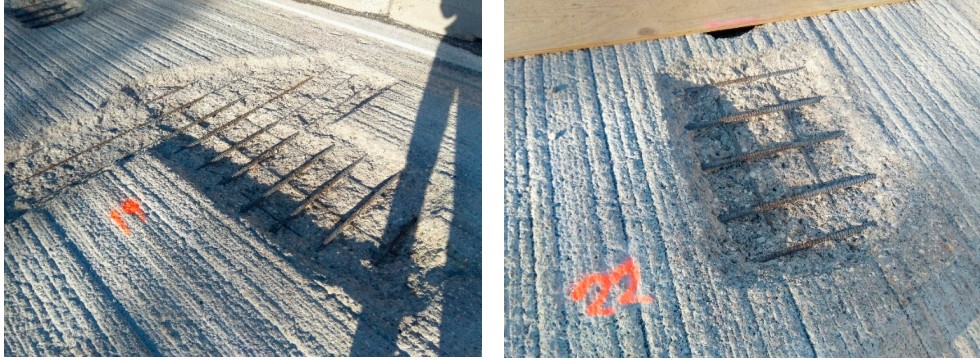

**Figure 19.** Images showing corroded parts of deck for validating GPR signals.

*3.6. Significance and Potential Use of Dataset*

Evaluating bridge decks using deep learning requires feature extraction from datasets for classification and detection of defects. The training dataset requires an adequately validated dataset for testing the model's performance. The dataset will provide an instrumental annotated dataset from three different NDE methods for training and validating AI models. Training datasets from bridges will help develop AI models for classifying bridge defects. Most of the existing pre-trained AI models are built with other image types.

The significance of SDNET2021 is highlighted as follows, providing data for in-service bridge decks. Most currently available NDE data have been generated from laboratory models and specimens.

- Developing a pre-trained model with annotated NDE dataset will be very useful in bridge evaluation. In addition, this dataset will provide a basis for developing pre-trained AI models for IE, GPR, and IRT datasets in classifying and detecting bridge defects.
- SDNET2021 also provides useful data for adopting data fusion in defect detection. Data fusion requires merging two or more NDE data to develop more accurate prediction and detection models. For example, the SDNET2021 dataset, which has been collected for IE, GPR, and IRT, could be fused to improve the detectability of defects compared to when adopted independently.
- IRT, IE, and GPR datasets have been annotated with validated ground truth. This dataset is a benchmark for evaluating bridge deck sub-surface defects.
- The dataset provides a means for continued concrete bridge deck evaluation with the aid of AI models, especially the use of convolutional neural network (CNN) models, which are still being explored. CNN use is promising for providing an unbiased and inexpensive way to analyze and interpret bridge evaluation data without operator input, compared to the conventional method of using expert evaluation.
- This reliable dataset will be available to professionals that need to investigate the relationships between concrete deck surfaces and subsurface defects using AI models.
- The dataset will be an excellent resource for developing data fusion of the different NDE data types, which will help professionals investigate the reliability and precision of one method relative to the other.
- A deep learning model trained on SDNET2021 can be used to investigate the detection of sub-surface delamination of varying sizes and depths.

## 4. Conclusions

The role of the validated and annotated dataset in AI is critical in benchmarking and developing effective and viable models with high accuracy for the detection of defects. To this end, the authors have developed a unique non-destructive evaluation dataset for subsurface defect detection in concrete bridge decks.

SDNET2021 contains 557 annotated (class 2 and 3 delaminated) and 1379 sound (class 1) IE signals, 214,943 annotated (class 2 and 3 delaminated) and 448,159 sound (class 1) annotated GPR signals and 1,718,083 annotated (class 2 and 3 delaminated) and 2,862,597 sound (class 1) pixels of annotated IRT images collected from five (5) in-service bridge decks in Grand Forks, ND, USA.

All NDE data were collected before the commencement of the bridge repairs. The data quality was evaluated using image quality metrics for images and signals. The evaluation indicated that the data presented in SDNET2021 were high-quality.

The dataset was validated and annotated using a set of ground truth maps obtained during repair works showing the class of deck removal. GPS location, size, and delaminated area removal were collected during the repairs. The ground truth was developed to show class 1 as Sound (No Delamination); class 2 as shallow delamination (delamination above the top bar mat), and class 3 as deeper delamination (delamination below the top bar mat).

Each delamination map showed the location and severity of subsurface damages which were used to classify and annotate SDNET2021.

Conventional techniques, such as image processing, frequency approach, and GPR profile (B-scan), were adopted for preliminary processing and initial validation of the infrared thermography, impact echo, and GPR datasets. The preliminary processing results indicated that the datasets and ground truth were reliable and ready for further processing and use.

SDNET2021 will contribute significantly to further studies for AI model development, allowing for the creation of models capable of delamination and defect classification and detection. The development of these models is essential for continued research in advanced NDE and structural health monitoring.

SDNET2021 will play a significant role in artificial intelligence development and benchmarking for NDE-based bridge deck evaluation. Significance results of SDNET2021 include:

- Providing data for in-service bridge decks,
- Benchmarking dataset for evaluating bridge deck sub-surface defects,
- Developing CNN for defect detection and classification,
- Being available to professionals for investigating the relationships between surfaces and subsurface defects using AI models,
- Developing data fusion of different NDE data types,
- Investigating the detection of sub-surface delamination of varying sizes and depths.

**Author Contributions:** Conceptualization, S.D.; methodology, S.D., E.I. and F.J.; software, E.I.; validation, E.I. and F.J.; formal analysis, E.I. and F.J.; investigation, E.I. and S.D.; resources, S.D.; data curation, S.D. and E.I.; writing—original draft preparation, E.I. and S.D.; writing—review and editing, E.I. and S.D.; visualization, E.I. and F.J.; supervision, S.D.; project administration, S.D.; funding acquisition, S.D. All authors have read and agreed to the published version of the manuscript.

**Funding:** This research was funded by the North Dakota Department of Transportation grant number 20.205.

**Data Availability Statement:** The data described in this paper are available under the University of North Dakota commons license. The raw IRT image dataset for the bridges is freely available on request to the corresponding author and NDDOT. The data described in this paper are published in the open-access UND repository, https://commons.und.edu/data/19 (Last accessed on 10 June 2022).

**Acknowledgments:** The authors would like to thank the North Dakota Department of Transportation for funding the data collection and other related activities performed in this study. The results of this study do not represent the official views of the North Dakota Department of Transportation. The authors would also express their gratitude to the following individuals who contributed to this investigation in data collection and management of the project: Jeff Cohen, BDI staff member; Sebastian Gomez, Mike Johnson, and Mitchell Banks (Skyscopes staff crew); Christman Caleb, Murphy Thomas, Johnson Michael (NDDOT project crew); Crowell Anna at the University of North Dakota for her great editorial works; and Amrita Das for her assistance during data annotation.

**Conflicts of Interest:** The authors declare no conflict of interest.

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
