# Peer review of "SDNET2021: Annotated NDE Dataset for Subsurface Structural Defects Detection in Concrete Bridge Decks"

_infrastructures, doi:10.3390/infrastructures7090107_

Round 1

Reviewer 1 Report

GENERAL OVERVIEW

The paper introduces the NDE dataset collected on real bridge structures, namely SDNET2021. The dataset contains data collected using very popular diagnostic methods, i.e., impact echo, infrared thermography, and ground penetrating radar. The considered data is described, processed, and discussed.

The topic of the presented paper meets the scope of the journal. The introduced dataset is valuable since the measurement were performed on existing bridge structures with real damage scenarios and it has the potential for further use. However, there are some issues that should be addressed before publication, mostly in the case of the presentation of results. The manuscript sometimes seems to be unfinished; it should be revised carefully in regard to the Journal’s requirements (template). In my opinion, the paper can be published in Infrastructures after a major revision, on the condition that Authors enhance the manuscript regarding given remarks. Some critical remarks and questions, that should be addressed, are presented below.

GENERAL REMARKS

1) Structure. The structure is logical, the following sections are organised reasonably.

2) Formatting. Compared to the Journal’s template, the formatting rules are sometimes not respected. Some flaws are mentioned in the following part of this review report.

3) Language, style, and formatting. In general, the language is good. However, there are many mistakes, e.g., typographic errors, lack of spaces between values and units, wrong font. To be sure of the quality of language, I suggest careful proofreading of the whole manuscript by a native speaker or using the professional proofreading service. Some of the flaws, but definitely not all are listed below:

– line 21 – there is a lack of space before ‘Each’,

– line 22 – doubled space should be deleted before ‘The NDE’, the same in line 52 after ‘Some’,

– line 25 – classes were written with capital letter before, however, ‘class 3’ is lower case here and in the following line, please unify here and throughout the manuscript,

– line 31 – the link is written in wrong font size; despite this, I am not convinced that links should be included in abstract,

– lines 105-110 – this sentence is long and does not have a good style; multiple repetition of ‘and’ is confusing; please, rewrite,

– line 124 – lack of space in ’142m’, the same error appears many times, e.g., in lines 155, 177, 179, 197, 200, 201, 204, 225 (in the table), 368,

– line 187 – typo in ‘se-up’,

– line 232 – the section title should be written in bold,

– line 237 – I do not find appropriate to write word ‘Classified’ with capital letter; the same in line 239,

– line 262 – section should have number, here, I suggest 4.1; similar in line 281 (4.2).

4) Introduction. The Introduction presents the background of the study in a satisfactory way.

5) Novelty. The novelty and the value of the current paper is appropriately emphasized.

6) Figures. The figures are mostly legible and have a good quality. However, they are sometimes not appropriately formatted in the case of font type and size. Some of the issues are mentioned below:

– Figure 1 caption – capital letters should be provided for ‘Google Earth’ and ‘Google Maps’,

– Figure 3b – there is no unit provided for the axes, please, add,

– Figure 4 – it is a bit confusing that the elements of subfigure 4a are marked as (a)-(f); this should be changed, e.g., to (1)-(6) or (A)-(F); the same in the description in lines 197-198,

– Figure 6 – it could be nice to distinguish areas of Class 2 and Class 3 with different colours to better visualize the differences,

– Figure 7a – it is not possible to distinguish signals for different classes; please change the line patterns or show these three signals in separate subfigures; according to x-axis, the notation ‘E^-5’ is not appropriate, please delete ‘^’,

– Figure 12 – there is no unit in the axes of both subfigures; the quality of subfigure 12b is low, the scaling of axes should be the same as in 12a (now 12b is unnaturally stretched),

– Figure 18 – what are symbols 2A, 3A, 1A etc.?

– Figure 19 – the caption should start with capital letter.

7) Tables. The tables are legible and well-present the data, however, some of the table captions are not appropriate: the caption should start with bold ‘Table X’ and end with full stop. What is more, the numbers of tables are repeated. Also, some particular remarks are given below:

– Table 2 – since date of inspection of all bridges is the same, it could be mentioned in the text, this information is redundant in the table,

– Table 7 – references are written with wrong font; what is more, after the names of the authors, the citations should be added using preferred style ‘[X]’,

– Table 5 (line 305) should be Table 8; the number of the following tables should be also updated,

– Table 6 (that should be Table 9, line 348) – since there are two scale factors a and b, which one is presented in the last column?

8) Equations. Equations are sometimes not appropriately formatted. Some issues are mentioned below:

– Eq. (1) – variables in equations should be written in italic to be distinguished with units; good example here: italic T is temperature, whereas standard T is tesla (unit); please reformat all equations,

– line 145 – while referring to variables from equations in the main text, they should be also written in italic,

– Eq. (2) and (3) – please change red colour of parentheses to black; additionally, functions like ‘sin’ or ‘cos’ should not be written in italic, on the contrary to variables T, R, S, X, Y, d, a, b, which should be also italic in the text (lines 343-345),

– Eq. (5)-(8) – the red lines should be changed to black, and the variables should be written in italic; the same in the text in lines 378-385.

9) References. The citation style is wrong, the references should be introduced in the text by numbers [X], not by name of the authors and year. The reference list should be organized in the order of appearance in the text. What is more, the font of reference list is wrong.

10) The manuscript lacks the type of paper above the title on a title page.

11) Line 3 – I do not find it appropriate to put degrees or title with the author’s name, please delete.

12) Lines 4-7 – since the affiliation of all authors is the same, the using of numbers is redundant; please use a single affiliation and give the authors’ emails in a single line with appropriate authors’ initials.

13) Lines 10-11 – I find the word count redundant, please delete.

14) Lines 12-13 – the abstract should start in the same line, in which word ‘Abstract’ is included.

15) Line 32 – keywords should be separated with semi-colon not comma.

16) Line 120 – since it is the first time when the bridges are mentioned, their names and symbols should be placed here.

17) Line 143 – what does ‘[ref?]’ mean here? Is it just a note that there should be some reference, but it was not added before manuscript submission? The same remark in lines: 191 and 196.

18) Line 192 – I find it discussible to write the specific limits for GPR antennas, e.g., there are GPR systems with 4 GHz antennas. Please delete these limits or replace them with more general phrases like ‘a few MHz’ and ‘a few GHz’.

19) The Author Contributions section should be rewritten to meet the requirements presented in the template. CRediT taxonomy should be used. It is also better to use authors’ initials instead of full names.

20) In the case of units, it is not appropriate to mix units from different systems, e.g., length is sometimes given in meters and sometimes in feet. Please, unify. What is more, a standard symbol for second is ‘s’ not ‘sec’ (line 187).

Reviewer 2 Report

The paper addresses the description of a new Non-Destructive Evaluation dataset collecting Infrared Thermography, Impact Echo, and Ground Penetrating Radar data collected from five in-service reinforced concrete bridge decks. A detailed description of the data collection for all the cases is provided, in addition to the information about the post-processing technique applied to the data. The quality of the collected data was verified and the data were properly annotated providing a distinct ground truth definition.

The authors present the data collection procedure fairly well, making the dataset valuable. In my opinion, the dataset presented is of scientific interest. However, there is no scientific contribution in terms of data analysis and interpretation through damage detection techniques. Moreover, the paper should be improved by being poorly written.

 Remarks:

There is no implementation of any damage detection strategy or algorithm, which might be use as a benchmark reference for future publications that will adopt this work. I think it would be interesting to implement an initial detection method to give a better understanding of the potential of the dataset.

1.     The objective of the paper does not emerge clearly from the abstract. The authors should revise it.

2.      The authors introduce directly the SDNET2021 dataset in the abstract without giving any type of context. The acronym is provided only later on in the introduction at line 105. Only by deduction, the reader understands that we are talking about a dataset. Not even the anacronym is explained. This goes in parallel with the first comment, the paper’s aim is not clear right away.

3.     The paper is poorly written. There are several grammar errors and inconsistencies all over the paper. The authors should revise it.

Here are some examples:

-        Missing a dot at line 21

-        At line 20 the term data is considered singular, while on line 22 is considered plural

-        The verb “exists” at line 47 should be “exist” since the subject is datasets and it is plural

-        Such as the term dataset at line 48 should be datasets (plural)

-        Missing commas on lines 116 and 117 and misposition of the dot

-        Sometimes you have closing dots in the subsection titles, sometimes not

-        There is a reference path missing at line 143

-        Missing references at 191

4.      The quality of the images is extremely poor (figure 9, 11, 12 a) or some of them are stretched (figure 12b)

Round 2

Reviewer 1 Report

In a nutshell, I am convinced that Authors appropriately addressed most of the remarks and made required changes. I list some still present flaws but those are only formatting ones, thus, if not corrected by Authors now, they will be eliminated during the production process. So, I am not hesitating to recommend accepting the paper.

– most of the equations are included as figures, which significantly deteriorate the quality; the same with some of the references to the variables in the main text,

– the numbers of equations should not be italic,

– the abbreviation for ‘equation’ is ‘eq.’ not ‘equ’,

– numbers in titles of sections 3 and 4 should change: ‘3.0’ and ‘4.0’ should just be ‘3.’ and ‘4.’,

– Figure 19 – the caption should start with capital letter,

– the style of references in reference list should be double checked.

Reviewer 2 Report

The authors addressed the required revisions.